



# Leveraging sap flow data in a catchment-scale hybrid model to improve soil moisture and transpiration estimates

Ralf Loritz[1], Maoya Bassiouni[2,3], Anke Hildebrandt[4,5,6], Sibylle K. Hassler[1,5], Erwin Zehe[1]

[1] Karlsruhe Institute of Technology (KIT), Institute of Water and River Basin Management - Hydrology, Karlsruhe, Germany
[2] Department of Crop Production Ecology, Swedish University of Agricultural Sciences, Uppsala, Sweden.
[3] Department of Environmental Science, Policy and Management, University of California, Berkeley, CA, USA
[4] Helmholtz Centre for Environmental Research – UFZ, Department Computational Hydrosystems, Leipzig, Germany
[5] Karlsruhe Institute of Technology (KIT), Institute of Meteorology and Climate Research – Atmospheric Trace Gases and Remote Sensing, Karlsruhe, Germany
[6] Friedrich Schiller University Jena, Institute of Geoscience, Jena, Germany

*Correspondence to*: Ralf Loritz ([Ralf.Loritz@kit.edu](Ralf.Loritz@kit.edu))

## Abstract

Sap flow encodes information about how plants regulate opening and closing of stomata in response to varying soil water supply and atmospheric water demand. This study leverages this valuable information with data-model integration and deep learning to estimate canopy conductance in a hybrid catchment-scale model for more accurate hydrological simulations. Using data from three consecutive growing seasons, we first highlight that integrating canopy conductance inferred from sap flow data in a hydrological model leads to more realistic soil moisture estimates than using the conventional Jarvis-Stewart equation, particularly during drought conditions. The applicability of this first approach is, however, limited to the period where sap flow data are available. To overcome this limitation, we subsequently train a deep learning network to predict catchment-averaged sap velocities based on standard hourly meteorological data. These simulated velocities are then used to estimate canopy conductance, allowing simulations for periods without sap flow data. We show that the hybrid model, which uses the canopy conductance from the machine learning approach, matches soil moisture and transpiration equally well as model runs using observed sap flow data and has good potential for extrapolation beyond the study site. We conclude that such hybrid approaches open promising perspectives for more parsimonious process parametrizations by improving our ability to incorporate novel or untypical data sets into hydrological models.

## 1 Introduction

Globally, about 26 to 40 % of the precipitation that falls on the continents is transpired by vegetation, making it one of the dominant fluxes of the terrestrial water cycle (*Dingman, 2015*). Seasonal variations in plant water use can thus significantly affect the water balance of catchments, modify its runoff generation, and change its dynamic water storage (*Brown et al., 2005;*



*Hrachowitz et al., 2021; Seibert et al., 2017*). Understanding the role of ecosystems in catchment hydrology is crucial, particularly when investigating the impacts of climate change (e.g. *Duethmann et al., 2020*). Estimating transpiration at the catchment scale is, however, challenging as plant water uptake is difficult to measure, parameterize and scale up from the individual plant to the ecosystem level (e.g. *Mencuccini et al., 2019*). As a consequence, the predictive performance of

hydrological models, which represent water balance and vegetation dynamics in a physically consisted manner, can be limited due to the a-priori chosen vegetation process parameterizations and parameter values (e.g. *Bennett and Nijssen, 2021; Gharari et al., 2021; Mendoza et al., 2015*). Improving these uncertain parameterizations requires methods that can combine process-based hydrological models with new information about how plant transpiration varies with environmental conditions.

Flux towers provide the state-of-the-art evapotranspiration data to train and validate hydrological models. One caveat in using these measurements is that they represent an effective flux integrating evaporation from the canopy interception store and the soil with plant transpiration. An accurate partitioning of this integral flux into its components is, however, of key importance for improving transpiration modelling under changing conditions (*Stoy et al., 2019*), including effects of land use changes such as deforestation *(e.g. Hrachowitz et al., 2021)* and forest regeneration (e.g. *Neill et al., 2021*). This is a key reason why sap

flow is used as independent measurement technique to characterize transpiration dynamics in forest (e.g. *Granier and Loustau, 1994*) and agriculture ecosystems (e.g. *Dugas et al., 1994*). While originally established in the plant physiology community, sap flow data have also proven useful in hydrological research. For instance, *Renner et al. (2016)* showed that stand composition of forests can counteract differences in sap flow on south and north facing slopes leading to similar transpiration rates on both expositions. *Hoek van Dijke et al. (2019)* found that the Normalized Difference Vegetation Index (NDVI)

successfully captured sap flow dynamics during the green-up phase, although it failed under dry conditions. *Hassler et al. (2018)* highlighted that spatial differences of atmospheric demands and soil moisture only explain a small fraction of observed spatial variation of sap flow, while site specific factors, like geology and aspect, were more important. These finding imply that accounting for relations between vegetation characteristics, hydro-meteorological drivers and catchment properties can improve transpiration estimates and exemplifies the potential of using sap flow data to advance hydrological simulations. The

value of sap flow information is emphasised by the growing availability of global open-source sap flow databases (*Poyatos et al., 2016*) that provides opportunities to develop generalized relations to better inform hydrological models at places where no sap flow data is available.

Plants adapt transpiration depending on atmospheric water demand and supply. One important regulation mechanism is the

opening and closing of the pores on their leaves, called stomata, to regulate their $CO_2$ and water vapour exchange with the atmosphere. This process crucially governs the transpiration of plants, which is also reflected by the wide range of stomatal conductance models that are available in hydrological models (e.g. *Damour et al., 2010*). One issue is that these stomatal conductance models typically rely on several site specific parameters and each approach has its own limitations, rendering the choice of the "*right*" process parameterization challenging. In this context it is interesting to note that sap flow can, besides





being used to estimate transpiration directly, also be used to infer canopy conductance or stomatal conductance scaled by leaf area index (LAI). This is done by inverting a simplified formulation of either Fick's Law or the Penman-Monteith equation (e.g. *Ewers and Oren, 2000; Köstner et al., 1992; Phillips and Oren, 1998*).

While the complex interactions between soil water supply, vegetation behaviour and meteorology are challenging to
parameterize in bottom-up empirical or physically based stomatal conductance models, machine learning methods have recently proven to be a particularly useful alternative to reproduce ecohydrological behaviour and estimate transpiration (e.g. *Fan et al., 2021; Zheng et al., 2021*). However, despite their recent success, machine learning approaches also have shortcomings as they do not ensure mass and energy conservation and lack physical constraints. The latter renders extrapolation and simulation under changing boundary conditions challenging. Hybrid models that combine physical knowledge of process
equations with the flexibility of data driven predictions are therefore a promising tool to estimate fluxes and state variable in the earth system (e.g. *Reichstein et al., 2019*).

In this study, we propose and test a hybrid machine learning approach to integrate sap flow data into process-based hydrological model, and explore opportunities for improving soil moisture and transpiration estimates at the catchment scale. Specifically,
we leverage an extensive sap flow dataset, spanning a drought period, in a sub catchment of the well-monitored and well-studied Attert experimental observatory (*Pfister et al., 2002*). We first integrate canopy conductance inferred from sap flow data into a process based hydrological model and compare its performance to the reference model that uses an empirical stomatal conductance equation. We then train a deep learning network based on standard hourly meteorological data, to predict sap flow beyond the temporal extent of the training period. These simulated velocities are then used to estimate canopy
conductance, allowing us to replace the empirical stomatal conductance equation in the hydrological model on forward simulations beyond the monitoring periods. Our results support the value of such hybrid model approaches by comparing the different model variants against each other and against hydrological data such as soil moisture and discharge. Importantly, we highlight the value of sap flow measurement campaigns for improving simulation at the catchment scale.

## 2 Materials and methods

### 2.1 Study area

The Weierbach is a 0.44 km$^2$ large experimental headwater catchment, nested in the Colpach catchment and located in Luxembourg (*Hissler et al., 2021*). The catchment is characterized by coarse-grained and highly permeable soils and a slate bedrock (Ardennes massif). The climate is temperate semi-marine, mean annual rainfall is 950 mm and mean monthly temperatures range between 0°C in January and 17°C in July. Precipitation is evenly distributed across the seasons while the
runoff generation has a distinct seasonal pattern with around 80% of the annual discharge is released between October and



March (*Loritz et al., 2021*). The Weierbach catchment is entirely forested and dominated (>70%) by deciduous beech trees (*Fagus sylvatica*) and oak trees (*Quercus spec*). A detailed description of the Weierbach catchment and a comprehensive open access hydrological data set can be found in *Hissler et al. (2021)*. The Colpach is the parenting catchment of the Weierbach, located in the same hydro-pedological area and characterized by a similar runoff generation and formation (*Loritz et al. 2019*), but it comprises a larger variety of land cover types (65 % forest, 35 % agriculture).

### 2.1.1 Hydro-meteorological data

This study requires hourly meteorological data to force the water balance simulations and to calculate canopy conductance. For all these purposes, we use data records from April 2014 to October 2016. We obtain air temperature (°C), relative humidity (%) and rainfall data (mm hr$^{-1}$) from the Holtz meteorological station available in the open-access dataset from *Hissler et al. (2021)*. We obtain wind speed (m s$^{-1}$) and global radiation (W m$^2$) measurements from a meteorological station around 500 m south-east of the catchment available from the Catchment as Organized Systems (CaOS) project observation network (*Zehe et al., 2014*). Additionally, we use discharge data and averaged soil moisture from *Hissler et al. (2021)* at 10 and 60 cm depth (based on six individual sensors in each depth) to quantify the performance of hydrological model simulations. Soil moisture was additionally corrected for a stone content of 10 and 30 % in 10 and 60 cm based on several soil profiles in the research area (Jackisch, 2015).

### 2.1.2 Sap velocity measurements

We use hourly sap velocities (cm h$^{-1}$), the rate of water flow through a tree, from three growing seasons (April – October; 2014 – 2016) of an extensive measurement campaign in the Colpach catchment (detailed description in *Hassler et al., 2018*). We use a subset of the original data set of *Hassler et al. (2018)* comprising 32 trees, including 17 beech trees (*Fagus sylvatica*), 11 oaks (*Quercus spec.*), 2 hornbeams (*Carpinus betulus*) and 2 common alders (*Alnus glutinosa*) with individual tree diameters at breast height ranging from 8 to 80 cm (average 32 cm). Sample distribution ranges from north to south facing slopes and up- and downslope sectors, specifically selected to capture the typical hydro-pedological characteristics of the Colpach and the Weierbach. The campaign equipped each tree before leaf out of the growing season with sap flow sensors, manufactured by East 30 (Washington, USA). The sensors have three measurement depths, at 5, 18 and 30 mm in the xylem and measure sap velocity with the heat ratio method (*Campbell et al., 1991, Burgess et al. 2001; Hassler et al 2018*). We estimate tree-specific sap velocities by calculating the median from the measurements at the three different xylem depths. We use the median to account for the skewed distribution of sap velocities inside the sap wood, as sap velocities typically decrease closer to the heartwood (e.g. *Gebauer et al. 2008, Jackisch et al. 2020*).





### 2.1.3 Catchment-level sap flow based transpiration

This study focuses on catchment-level transpiration to circumvent the challenge and uncertainty of characterizing transpiration from individual tree sap flow (e.g. *Gebauer et al., 2008; Zhang et al., 2015*) and to remain scale consistent with simulated transpiration of the hydrological model. We employ an integral approach, assuming that the tree sample is representative for the age spectrum in the catchment and that trees dominate transpiration in this forested catchment compared to understory and herbaceous vegetation. We average the 32 tree-specific sap velocities to obtain a time series representing an average tree in

the study area. We then obtain average hourly catchment-level sap flow based transpiration per unit ground area ($T_{sap}$, m s$^{-1}$) by multiplying the catchment-averaged sap velocity by the catchment-averaged tree density of 42 m$^2$ ha$^{-1}$ (*Hassler et al 2018*). This calculation assumes that water storage in the tree relative to the transpiration flux is negligible. Therefore, the observed daytime water flux through the tree is equal to the transpiration flux through the leaves into the atmosphere, with negligible time lags between dynamics of sap flow (converted to $T_{sap}$) and environmental variables (*Tyree and Ewers, 1991*). We use

$T_{sap}$ data to derive observation-based canopy conductance estimates and to evaluate model simulations.

### 2.2 Hydrological model CATFLOW

  We model the water balance of the Weierbach with CATFLOW (*Maurer, 1997; Zehe et al., 2001*), a process-based hydrological model. CATFLOW discretizes hillslopes along a 2-dimensional cross-section using curvilinear orthogonal coordinates and a storage weighting function to represent the varying hillslope width. The model simulates soil water dynamics

based on the Darcy-Richards equation and represents surface runoff by a diffusion wave approximation of the Saint-Venant equation. CATFLOW estimates three components of the evapotranspiration flux per unit ground area, namely 1) direct evaporation of canopy interception, 2) transpiration from canopy leaves and 3) soil water evaporation, separately with a surface energy balance approach using the Penman-Monteith equation. For each component, soil, canopy (section 2.2.2) and canopy interception conductances are each parameterized differently with a set of empirical equations. Additional CATFLOW model

descriptions can be found in *Loritz et al. (2021)* and in *Loritz et al. (2017)*.

### 2.2.1 CATFLOW implementation of three canopy conductance variations

  We implement three approaches to estimate canopy conductance in the Penman-Monteith equation for transpiration in CATFLOW. The *reference model* implements canopy conductance calculated by the empirical *Jarvis–Stewart* equation, which is the built-in stomatal conductance equation of CATFLOW ($g_c$*Jarvis*; section 2.2.2) scaled by the LAI. The second model

variant is a *model-data integration*, which implements canopy conductance based on hourly observed sap flow data for all three growing seasons from 2014 to 2016 ($g_c$*sap;* section 2.2.3). The third model variant is a *hybrid model*, which implements canopy conductance based on sap flow predictions from a deep learning network ($g_c$*DL;* section 2.2.4).

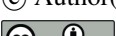



### 2.2.2 Reference model: Canopy conductance from the reference empirical canopy conductance equation ($g_c Jarvis$)

The *Jarvis–Stewart* model (*Jarvis, 1976; Stewart, 1988*) is a widely applied empirical equation for stomatal conductance as a function of plant available radiation (W m$^{-2}$), vapour pressure deficit (Pa), temperature (°C), matric water potential of the soil (m), and is implemented in CATFLOW. The canopy conductance per unit ground area ($g_c Jarvis$) is calculated from the leaf-level stomatal conductance scaled by leaf area index (LAI, m$^2$ leaf m$^{-2}$ ground). Parameters of the *Jarvis–Stewart* model are prescribed according to a lookup table and are based on mean parameter values (e.g. *rooting depth*, *plant albedo*, *interception*

*capacity*, etc.) for beech trees taken from *Breuer et al. (2003)*. LAI measurements are taken from satellites observations and change daily (Visible Infrared Imaging Radiometer Suite; VIIRIS). The model variant that uses $g_c Jarvis$ to estimate transpiration serves as *reference model* in this study.

### 2.2.3 Model-data integration: Canopy conductance from sap velocity measurements ($g_c sap$)

We use a big-leaf approach, in line with most catchment-scale transpiration models, to infer conductance to water vapour per

unit ground area ($g_c sap$; m s$^{-1}$) from sap velocity and meteorological data (wind speed, air temperature, and relative humidity). We assume a well-mixed, convective boundary layer during daytime, with high wind speed, small leaves, and similar leaf and air temperature. Given these common simplifying assumptions (e.g. E*wers and Oren, 2000; Köstner et al., 1992*), we neglect leaf boundary layer conductance and approximate the difference in water vapour concentration driving the vapour diffusion through the saturated air space in the leaves to the atmosphere by the air vapour pressure deficit ($e_s - e_a$; Pa). Hence, we can

invert Fick's Law following *Monteith and Unsworth (2013)* to calculate total water vapour conductance $g_t sap$ (m s$^{-1}$) as:

$$g_t sap = \frac{\gamma \lambda}{C_p \rho (e_s - e_a)} T_{sap} \qquad (1)$$

where $\gamma$ is the psychometric constant (Pa K$^{-1}$); $\lambda$ is the latent heat of vaporization of water (MJ kg$^{-1}$); $C_p$ is the specific heat of air (J kg$^{-1}$ K$^{-1}$); $\rho$ is air density (kg m$^{-3}$); $\gamma, \lambda, C_p, \rho$ are all a function of air temperature; and $T_{sap}$ (m s$^{-1}$) is the average catchment transpiration rate derived from sap velocities.


The total conductance $g_t sap$ represents the series of both $g_c sap$ and the aerodynamic conductance ($g_a$, m s$^{-1}$). The latter is estimated from wind speed and canopy height following the FAO reference approach (*Allen et al. 1998*). Finally, we obtain the time series of canopy conductance $g_c sap$ inferred from sap velocities as:

$$\frac{1}{g_c sap} = \frac{1}{g_t sap} - \frac{1}{g_a} \qquad (2)$$





This big leaf approach assumes that all canopy leaves in the catchment respond to the same environmental conditions and behave in the same way. This is reasonable, because hydro-meteorological data explained only a small fraction of spatial variability in sap flow velocities in the study site (*Hassler et al 2018*).

We implement canopy conductance inferred from observed and simulated sap velocities ($g_c sap$, $g_c DL$ explained in section
2.2.4) in CATFLOW only during the time steps for which the assumptions of Eq (1) are met (*Köstner et al., 1992*; *Phillips and Oren, 1998*): dry canopy (canopy interception storage < 0 mm); daytime (between 6:00 and 22:00); well-mixed atmosphere ($\frac{1}{g_a}$ is at least 10 s m$^{-1}$ larger than $\frac{1}{g_t}$); air vapour pressure deficit > 100 Pa. When these conditions are not met, the transpiration flux and stomatal conductance are generally low and we fill in the gaps with canopy conductance estimates from the built-in *Jarvis–Stewart* model. We smooth canopy conductance time series inferred from observed and predicted sap velocities using
a rolling mean with a three hour window that uses the three previous time steps to allow forward simulations.

### 2.2.4 Hybrid model: Canopy conductance from deep learning based sap flow predictions ($gcDL$)

We train a deep learning model to estimate hourly sap flow using the 2014 and 2016 data for training and the growing season of 2015 for testing. We choose the 2015 growing season as the test period because it has been identified as a drought year, during which transpiration was impacted by plant water stress (*Hoek van Dijke et al., 2019*). The deep learning network is
driven by the same hourly meteorological inputs as catchment models (temperature, relative humidity, global radiation, rainfall and wind speed). Additionally, the deep learning uses a sequence length of 96 (lag time of 96 hours preceding the prediction time step). Input and target features are standardized by subtracting their means and scaling by their standard deviations. The network consists of four layers (input, two hidden, output) with 128 cell/hidden states. The input and first hidden layer of the network use gated recurred units (GRUs); are followed by a second hidden linear layer with a relu activation function; finally
the output is a linear layer without an activation function. Between each layer we add 10 % dropout to avoid overfitting to the training data (regularization). The structure of the network is identified by trial and error. We use the mean-square error as loss function, train the model in 500 epochs with a batch size of 360 and report the root mean square error (rmse) in the results. We use an ADAM optimizer with a fixed learning rate schedule. The initial learning rate is set at 1e-3 and decreases by a factor of 0.95 after each epoch. Additionally, after the 400 epoch we use a stochastic weight averaging (SWU) approach with a learning
rate of 0.001 to improve the ability of the network to generalize in comparison to using exclusively an ADAM optimizer. We use the simulated sap flow velocities to estimate $g_c DL$ using the same method and under the same environmental condition as applied to estimate $g_c sap$ (Eq. 1-2).

### 2.2.5 CATFLOW parameterization

We use the well-tested, representative hillslope model from *Loritz et al. (2017, 2021)* to simulate the water balance of the
Weierbach using CATFLOW. The representative hillslope model was setup based on field data for the bedrock topography,

soil properties and surface topography. The model was fine-tuned by exclusively adjusting the spatially explicit macropore network (approach described in detail in *Wienhöfer and Zehe, 2014*) with the goal of matching the seasonal water balance and the hydrograph of the parenting Colpach catchment during the hydrological year October 2013 to October 2014. *Loritz et al. (2017)* showed that the representative hillslope model predicts the hydrograph of the Weierbach with a Nash-Sutcliff efficiency

(NSE) of ≈ 0.7 and a Kling Gupta efficiency of ≈ 0.8 for the hydrological year 2012/13 (test period) and the hydrological year 2013/14 (training period) individually.

The simulation period in this study starts on the 1st of April 2014 and runs until 31th October 2016. This is preceded by a model spin-up starting in October 2013 with initial states of 70% volumetric water content. We are using the exact same

parameterization as explained in detail in our previous studies (*Loritz et al., 2017, 2021)* and do no re-calibration of any model parameters besides changes described above to estimate the canopy conductance.

## 3 Results

### 3.1 Sap flow data-model integration provides realistic canopy conductance and water balance estimates for a temperate beech forest

The daily averaged canopy conductance (m s$^{-1}$) inferred from the sap flow measurements ($g_csap$) and those estimated by the a-priori parameterized CATFLOW built-in stomatal conductance equation ($g_cJarvis$) correlate well, although $g_cJarvis$ estimates are on average lower (Fig. 1a). Regardless, the spearman rank correlation between $g_cJarvis$ and $g_csap$ is 0.85 and the rmse is 0.01 m s$^{-1}$. The $g_csap$ estimates are within a reasonable range for a beech-dominated temperate forests and comparable to literature values using a similar approach (inverse Penman-Monteith equation) based on six beech trees in the Czech

Republic (*Su et al. 2019*). Differences between $g_cJarvis$ and $g_csap$ are also reflected, although weaker, in the monthly transpiration estimates (Fig. 1 b). The CATFLOW model variant using $g_csap$ (model-data integration) estimates about 130 mm more transpiration compared to the reference model variant using $g_cJarvis$ for all three hydrological years, with the largest monthly difference of 21 mm month$^{-1}$ in May 2015 (31 mm of total rainfall in May 2015).

Implementing $g_csap$ instead of $g_cJarvis$ in CATFLOW has only a weak effect on simulated runoff with a slight decline of the NSE from 0.75 to 0.71 over the three-year period. This decrease in predictive performance likely occurs because the macropore network was tuned to optimize the streamflow of the Weierbach with $g_cJarvis$ and not $g_csap$. This entails that a better performance could likely be achieved by tuning the macropore network once more with $g_csap$. However, we do not to perform further CATFLOW calibrations because our goal is to demonstrate the value of sap flow data in improving transpiration and

soil moisture estimates and do not aim to obtain the highest performance in streamflow simulation.



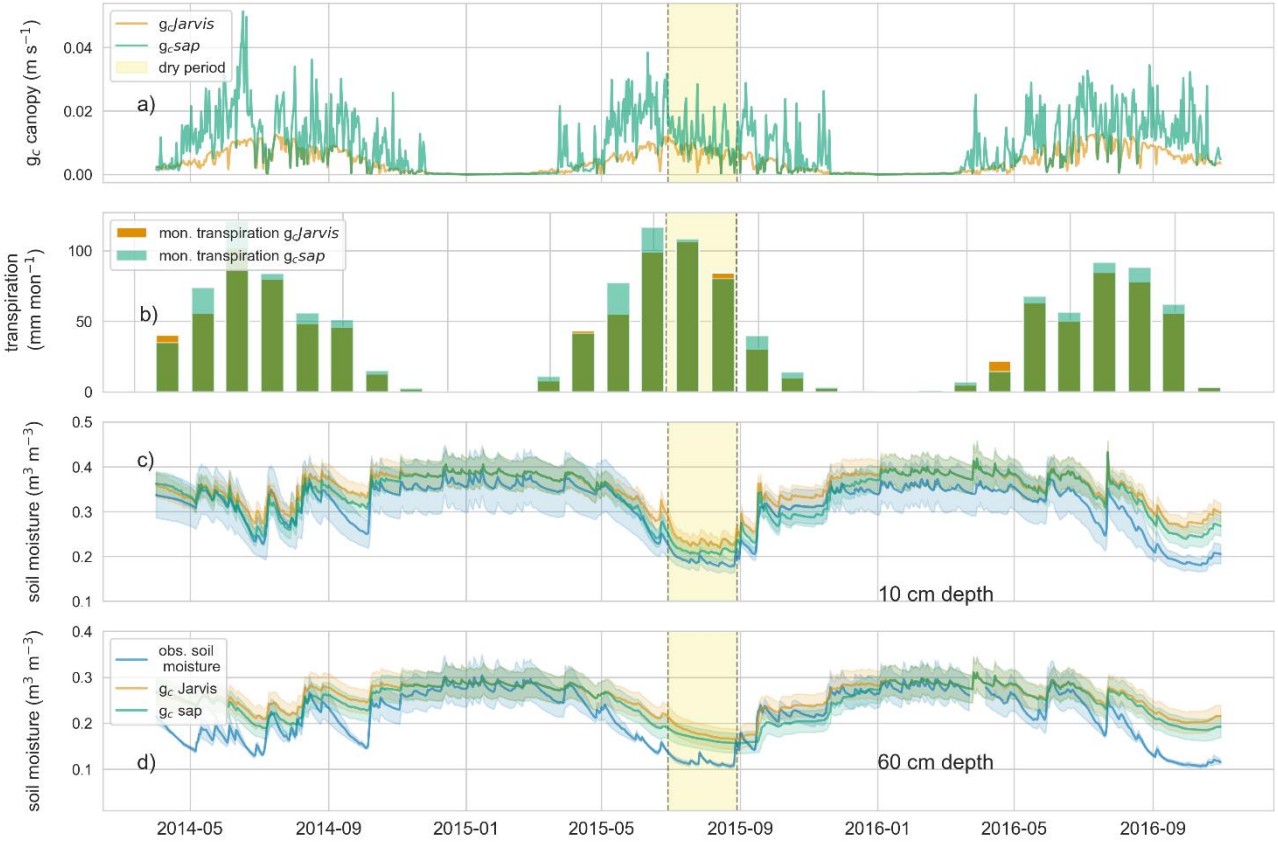

**Figure 1. a) daily averaged canopy conductance estimates for $g_csap$ (green) and $g_cJarvis$ (orange); b) monthly transpiration sums estimated using $g_csap$ (green) and $g_cJarvis$ (orange); observed (blue) and simulated soil moisture ± standard deviation of the corresponding simulation and observations ($g_csap$: green; $g_cJarvis$: orange) at 10 (c) and 60 cm depth (d). Highlighted in yellow is a dry period from July to August 2015.**

## 3.2 Ecohydrological simulations differ most during drought periods

Noticeable ecohydrological relevant model improvements using $g_csap$ occur during drought periods. The Weierbach fell dry on 61 days (> 0.001 mm h$^{-1}$) during the three year record. This period is only slightly overestimated by CATFLOW using $g_csap$ (63 days), while it is substantially underestimated using the reference model with $g_cJarvis$ (46 days). Both model variants ($g_cJarvis$ and $g_csap$) correlate well with the observed soil moisture in 10 and 60 cm with Spearman rank coefficients of around 0.9. However, simulations using $g_csap$ result in overall lower soil moisture values with the largest difference in October 2015 (Fig. 1 a and b). Using $g_csap$ instead of $g_cJarvis$ reduces the rmse in the 2015 growing season from 0.033 to 0.01 (0.046 to 0.034) m$^3$ m$^{-3}$ at 10 (and 60) cm depth. Furthermore, using $g_csap$ instead of $g_cJarvis$ leads to an average of about 2 mm less catchment storage after each of the three growing seasons. These storage differences are almost completely recharged in winter, typically until January, due to the wet autumns in the region. However, after the three growing seasons, the bedrock water





storage (characterized by very low hydraulic conductivities and low porosities) is on average 2 to 4 % lower when using $g_c sap$ compared to $g_c Jarvis$ after three years of simulations.

## 3.3 Deep learning accurately extrapolates sap flow data to different time periods and locations

Fig.2 a displays hourly simulated sap flow (cm h$^{-1}$) estimated by the deep learning model against observed sap flow (cm h$^{-1}$)

at daytime (6:00 and 22:00) of the growing season 2015 (test period). Simulated sap flow differs from observed sap flow by an rmse of 0.8 cm h$^{-1}$ during the training period (growing seasons 2014 and 2016) and 1.1 cm h$^{-1}$ during testing period. The spearman rank correlation between the observed and simulated sap flow in the test period is 0.91, indicating the ability of the deep learning model to capture the general dynamics of sap flow using hourly meteorological data as predictors. Sap flow during the dry spell in July and August 2015 is on average overestimated by the deep learning model. However, when adding

randomly picked 15 continuous days of the dry period to the training sample (and removing those from the test sample) this bias and the rmse are significantly reduced to 0.85 cm h$^{-1}$. Furthermore, we also tested the ability of the deep learning network to predict sap flow in a nearby catchment with a different geological and pedological setting but similar forest landcover. This first test suggests that the deep learning network can predict sap flow also in this test catchment even with lower errors as in the training catchment. This good out of sample performance points to the algorithm's ability to also extrapolate to higher

unseen sap flows without further training (Appendix A1).

## 3.4 The hybrid model provides accurate canopy conductance and water balance estimates

The canopy conductance inferred from the observed sap flow ($g_c sap$) and based on the simulated sap flow ($g_c DL$) are compared in Fig.2 b. The two estimates differ by a rmse of 0.01 m s$^{-1}$ in the test period and have a Spearman rank correlation of 0.9. The relation between the conductance estimates based on observed, $g_c sap$, and simulated sap flow, $g_c DL$; is characterized by more

and stronger outliers (residual larger than 0.025 m s$^{-1}$, Fig.2 b). Note that more than 75 % of these outliers occur in the morning (6:00 to 10:00) or evening time (16:00 to 22:00). During these times, the Fick's law approximation is very sensitive to little changes in sap velocities but transpiration is typically very low during these periods. This is further underpinned by the comparison of monthly transpiration sums displayed in Fig. 2 c. The differences in using $g_c sap$ or $g_c DL$ are less than 3 mm month$^{-1}$ during the majority of the growing season 2015 and increase only in July and August to 7 and 9 mm month$^{-1}$. In this

period, sap flow and to a smaller extent the corresponding $g_c$ values are systematically overestimated by the deep learning network (Fig.2 a). As stated above, adding 15 dry days to the training data can reduce these biases and decrease the transpiration differences in July and August below 4 mm month$^{-1}$. However, even without changing the training data of the deep learning network, the effect on simulated soil moisture dynamics is minor (Fig. 2 d). This is because the $g_c DL$ based model slightly underestimates transpiration in May and June, which is then compensated in July and August and the simulated soil moisture

from $g_c DL$ and $g_c sap$ differ only by a rmse of 0.003 m$^3$ m$^{-3}$ in 20 and 0.002 m$^3$ m$^{-3}$ in 40 cm from 1$^{st}$ May 2015 to 31$^{th}$October 2015.





## 3.5 The hybrid model improves the diurnal cycle of canopy conductance compared to the reference model

Fig. 3 shows three diurnal cycles of $g_cJarvis$, $g_csap$ and $g_cDL$ in June, July and August. $g_csap$ is about twice as high in June compared to August and shows a stronger decline in conductance during midday in July and August. While such patterns are typical for humid forests under dry conditions (*Su et al., 2019*), they are not or only weakly captured by the *Jarvis-Stewart* model ($g_cJarvis$), which suggests a relatively constant conductance during day time. As already indicated by the high correlation between $g_cDL$ and $g_csap$, the former also captures the dynamics of the diurnal cycles well. However, the $g_cDL$ model under- or overestimates several peaks, particular during the morning and evening hours. This is in line with Fig. 2b and

explains the larger spread of the $g_c$ estimates in contrast to sap flow predictions. The absolute cumulated difference of the transpiration estimates using either $g_cDL$ or $g_csap$ in the chosen three-day period is with 0.01, 0.014 and 0.07 mm day$^{-1}$ low and highlights that errors in $g_c$ estimates in the morning and evening are less important for transpiration estimates.

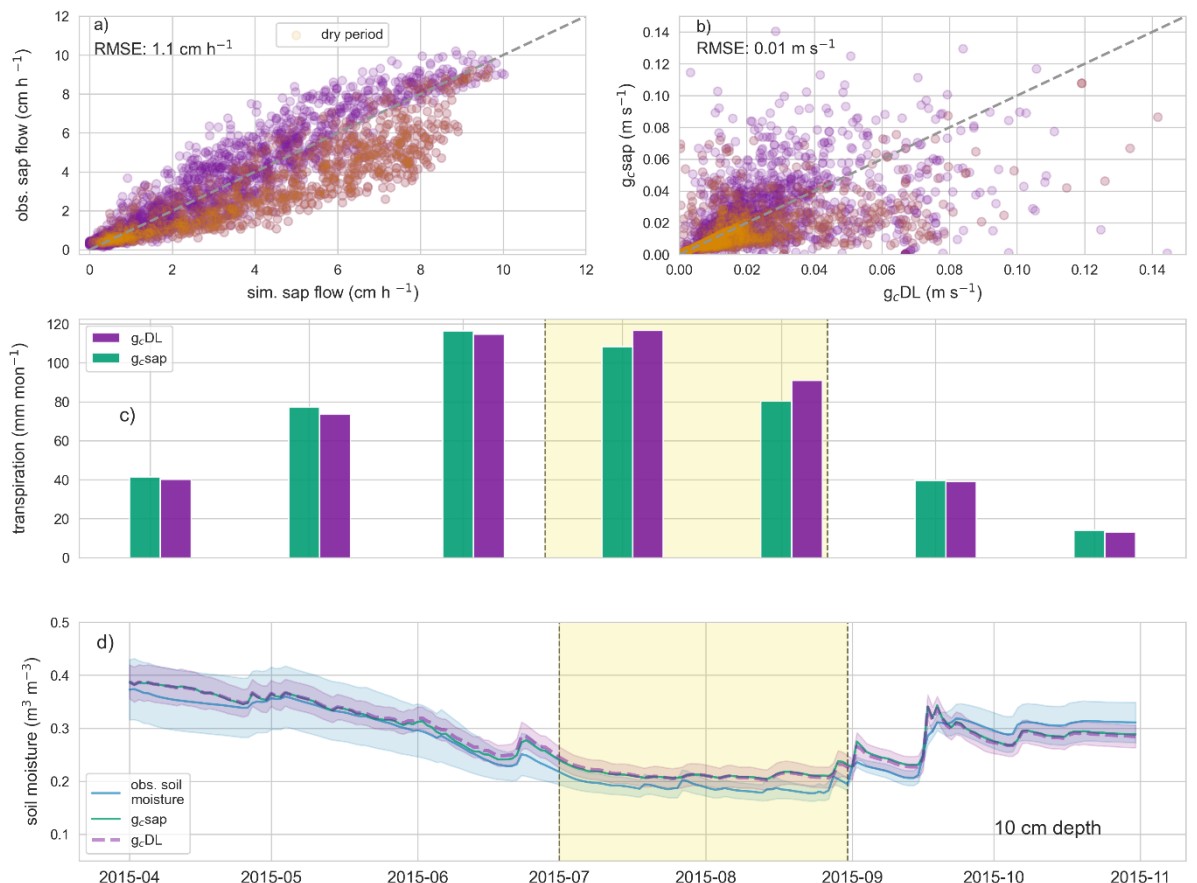

**Figure 2. a) hourly observed catchment-averaged sap flow and simulated sap flow in the growing season 2015; b) hourly canopy conductances based on the hourly observed sap flow ($g_csap$) and simulated sap flow ($g_cDL$); orange points in a and b are simulations**





**or observations within the dry period of July and August 2015; c) monthly transpiration sums estimated by $g_csap$ (green) and $gcDL$ (purple); d) observed (blue) and simulated soil moisture ($g_csap$: green; $gcDL$: purple) in 20 cm. Highlighted in yellow is a dry period from July to August in the growing season 2015.**

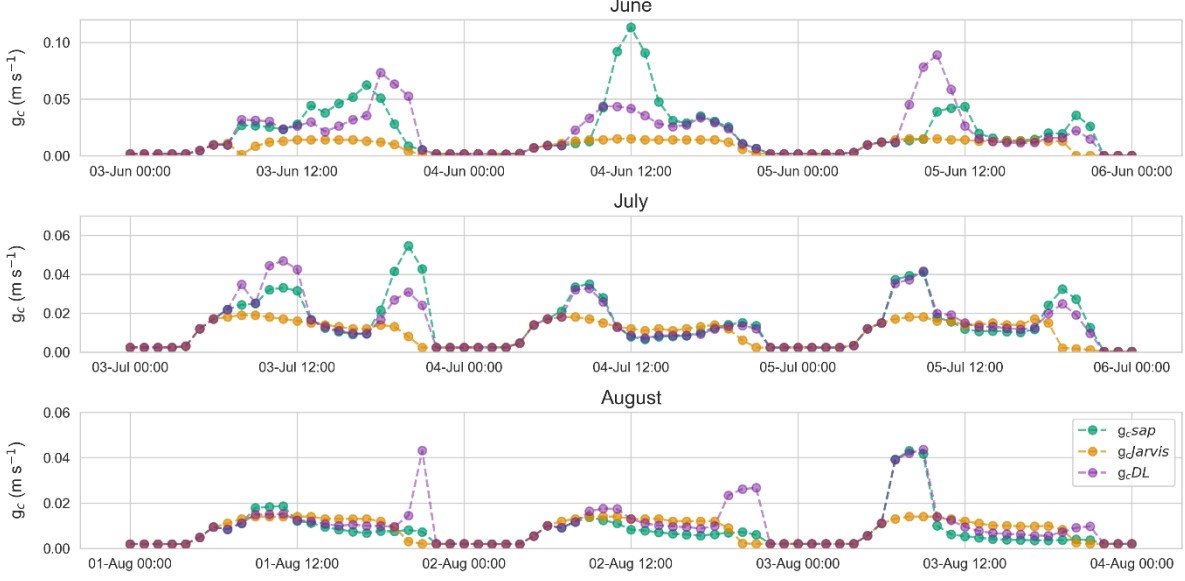


**Figure 3. Hourly canopy conductances of $g_csap$ (green), $g_cJarvis$ (orange) and $g_cDL$ (purple) for three selected days in June, July and August in the growing season 2015.**

# 4 Discussion

## 4.1 Integrating sap flow data in a catchment-scale hydrological model

The comparison between both stomatal conductance models revealed that the a-priori parameterized *Jarvis-Stewart* model (*Jarvis, 1976; Stewart, 1988*), in combination with the satellites based VIIRIS LAI values, clearly underestimated the canopy conductance, particularly during the spring and early summer. This bias could potentially be corrected by tuning the parameters of the *Jarvis–Stewart* equation. However, beyond revealing absolute errors in the seasonal cycle, the sap flow based stomatal conductance model also demonstrates that the *Jarvis–Stewart* model is not able to reproduce diurnal hydraulic feedbacks

reflected in the dips in canopy conductance during the mid-day water stress period. Simulating these dips would, most likely, require a more complex stomatal conductance model. On the other hand theses dynamics are embedded in the sap flow data and were well reproduced by the deep learning network. Therefore, learning this information from sap flow data with a deep learning network provides an avenue for catchment models to reproduce plant hydraulic behaviour without explicitly parameterizing the soil-plant-atmosphere continuum at the catchment scale, which is complex and uncertain (*Mencuccini et*

*al., 2019).*





Our results go beyond the established approach of estimating canopy conductance from sap flow data by directly integrating the data in a catchment-scale hydrological model and improving water balance simulations. Additionally, we can demonstrate the value of sap flow data in identifying suitable catchment-specific model parameterizations (*Gupta et al., 1999*) and show how the stomatal conductance model can be replaced by a data model integration. Using the sap flow to calculate canopy

conductance instead of transpiration has thereby the advantage of omitting species-dependent errors in estimating the sap wood area and sap velocity distributions within the xylem. Faulty estimates of these parameters can lead to an overestimation of daily water use of up to 78 % for oak trees and -42 % in case of oriental arborvitae trees as shown by *Zhang et al. (2015)*. Nevertheless, the results of the deep learning network underpin the possibility to predict sap flow with a machine learning approach. This approach could then be extended to estimate transpiration based on catchment averaged species dependent

parameters, which could, for instance, be estimated by LiDAR measurements (*Fassnacht et al., 2016*).

## 4.2 Predicting canopy conductance using sap flow and a deep learning network

Recent studies have shown the large potential of decision tree based machine learning algorithms for ecohydrological applications with a focus on predicting sap flow *(Ellsäßer et al., 2020)* or stomata conductances *(Saunders et al., 2021)* using meteorological data. In this study, we showed that deep learning networks are also suitable tools to predict sap flow by

exclusively using meteorological variables as input. Only during the dry period in the growing season 2015 where the dormant trees most likely experienced water stress (*Hoek van Dijke et al., 2019*) did the deep learning network systematically overestimate sap flow. The latter was the reason to choose 2015 as test period and not 2016, which would have kept the chronological order and led to overall lower errors without bias. Initial tests reveal that adding randomly picked 15 continuous days during the drought period to the model training can reduce the residuals as well as the bias significantly, although soil

moisture data were still not included as input. This indicates the potential of the deep learning networks to mimic sap flow also under water stress and solely based on meteorological input. The latter entails that the information about the drought period is already within the meteorological input and different aggregations and combinations of the input variables, for instance, by estimating drought indices like the standardized precipitation index (SPI) could potentially further improve the prediction of sap flow under limited water availability. This study highlights thereby the potential of the introduced deep learning approach,

but a more systematic investigation is required. Specifically, a next step could be to explore the potential of implementing the deep learning network such that the internal hydrological model states (especially soil water status) affect the sap flow predictions and the corresponding conductances. A similar hybrid modelling approach has, lately shown large potential to represent turbulent heat fluxes in hydrological models (Bennett and Nijssen, 2021).

## 4.3 Generalizing canopy conductance models based on sap flow data

This study is based on an unique data set with several sap flow sensors installed in different trees and locations as well as over several growing seasons (*Hassler et al., 2018*). Such data sets are labour intensive and rare although sap flow monitoring has





become more common. While our proof of concept is limited to well-monitored experimental catchments, initial tests show that the deep learning network is capable of reproducing sap flow in a neighbouring catchment, characterised by a similar forest structure but different hydro-pedological setting, even with lower residuals (Appendix A1). Approaches like transfer

leaning, a concept to pre-train layers in a deep learning network on a large data set and only fine tune a subset of these layers in the destination area, might be used to predict sap flow also in a catchment with very little sap flow data available. Additionally, global and open data sets like SAPFLUXNET (*Poyatos et al., 2016*) in combination with catchment or forest properties offer opportunities to generalize our proposed approach .While machine learning predictions cannot directly advance understanding of the soil-plant-atmosphere continuum, we nevertheless show that they can be an improvement

compared to reference empirical models that, if ill parameterized (e.g. *Mencuccini et al., 2019; Mendoza et al., 2015*), are known to poorly capture non-linear responses of plant water stress at the seasonal and diurnal time scales. Using machine learning sap flow predictions in combination with the inversed Fick's law offers hence the possibility to replace stomatal conductance models entirely in hydrological models.

## 5 Conclusion

The main findings from our study leveraging sap flow data in a catchment-scale model are as follows:

1. Hourly, catchment averaged sap flow can be used to estimate canopy conductance and inform a process based hydrological catchment model to improve soil moisture and transpiration estimates.

2. Seasonal and diurnal model improvements were notable during drought periods when the reference empirical model underestimated plant water stress and point to the valuable ecohydrological information encoded in sap

flow data.

3. Deep learning networks are suitable tools to predict sap flow by exclusively using meteorological variables as input and offer promising avenues for developing generalized canopy conductance models for forward simulations beyond the monitoring time period and catchment location.

This study highlights the potential of sap flow data for improving hydrological simulations at the catchment scale by either constraining or informing hydrological models. We argue that sap flow sensors measure crucial information about one of the major fluxes of the hydrological cycle and should become the norm in experimental hydrology as soil moisture sensors, piezometer or gauging station are already today.





*Code and data availability.* All simulation results and the catchment averaged sap flow data are available from the lead author on request and will be made public (e.g. Zenodo) in case the MS is accepted for publication. The entire sap flow data set is available by SH on request.

*Author contributions.* RL and MB designed the study and wrote the paper. RL carried out all analysis and model simulations.
SH contributed expertise about sap flow measurements and provided the quality-controlled sap flow data. AH and EZ contributed to interpreting results and editing the paper.

*Competing interests.* EZ and AH are members of the editorial board of Hydrology and Earth System Sciences.

*Acknowledgements.* This research contributes to the Catchments As Organized Systems (CAOS) research group (FOR 1598), funded by the German Science Foundation (DFG ZE 533/11-1, ZE 533/12-1). MB received funding from the European Commission and Swedish Research Council for Sustainable Development (FORMAS) (grant 2018-02787) in the frame of the international consortium iAqueduct financed under the 2018 Joint call of the WaterWorks2017 ERA-NET Cofund.





## Appendix

### A1 Sap flow predictions in Huewelerbach

The Huewelerbach is a 2.7 km$^2$ large headwater catchment located in Luxembourg within the experimental Attert basin (*Pfister et al., 2002*). The prevailing geology is sandstones above an impermeable layer of clay stones. The climate is temperate semi-oceanic, mean annual rainfall is 845 mm (*Pfister et al., 2017*) and mean monthly temperatures range between 0°C in January and 17°C in July. The catchment is entirely forested and dominated by deciduous beech trees. Meteorological data to run the deep learning network in this Appendix consisted of hourly global radiation (W m$^2$), temperature (°C), wind speed (m s$^{-1}$) and relative humidity (%). Temperature and relative humidity are measured at a meteorological station located 3 km south of the catchment from a station operated by the "Administration des Services Techniques de l'Agriculture" (ASTA). Wind speed and global radiation are measured at a meteorological station in close proximity of the catchment that belonged to the CAOS Project observation network.

We use sap flow velocities from one growing seasons (April – October 2015) measured within or in close proximity to the Huewelerbach catchment. Tree species consist of 27 beech trees (Fagus sylvatica) 7 Oaks (Quercus spec.), and 2 hornbeams (Carpinus betulus) with individual tree diameter at breast height ranging from 22 to 91 cm (average 53 cm ). Sap flow was measured and aggregated similarly as described in the method section.

Fig. A1 shows the simulated and observed hourly sap flow in the Weierbach and Huewelerbach for the growing season 2015. Sap flow was predicted using the same deep learning network trained exclusively in the Weierbach (growing season 2014 and 2016). There was no further change to that network. The deep learning network was capable of predicting sap flow in the Huewelerbach in better agreement with the observations than in the training catchment. One main reason for this performance increase is that although in close proximity to the Weierbach the dormant trees in the Huewelerbach did not experience water stress in 2015 most likely due to a large and accessible groundwater store (*Hoek van Dijke et al., 2019*). Other factors including, as higher quality meteorological data or (potentially) sap flow data might also play a role but were not further investigated. Interestingly, the deep learning network is capable to simulate overall higher sap flow in the Huewelerbach although such values have not been observed in the Weierbach. This supports the ability of the deep learning model to extrapolate in different sites.



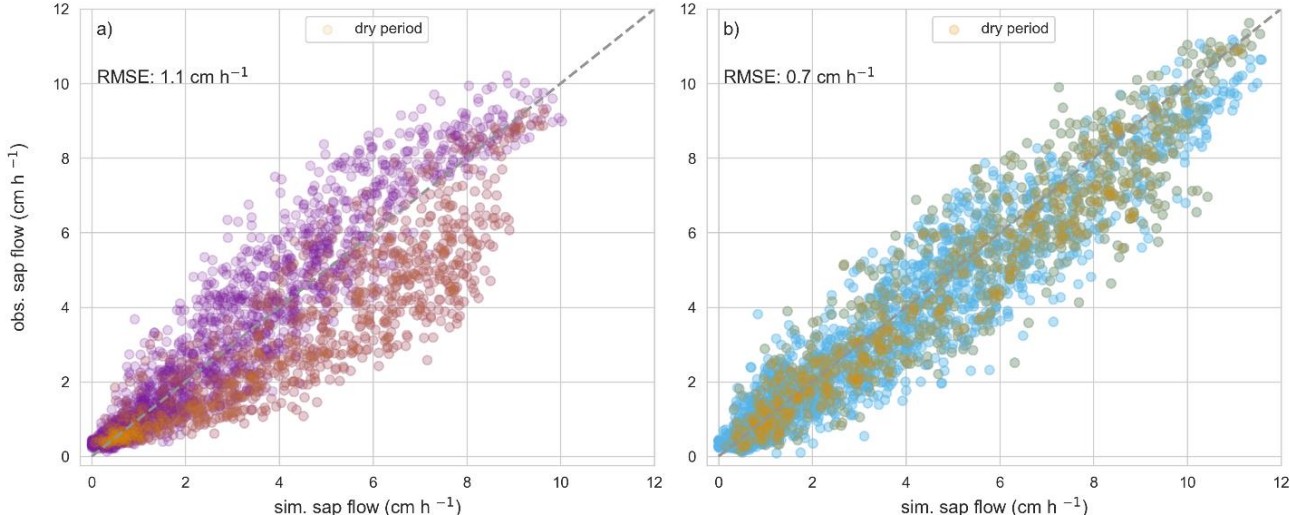

**Figure A1 a) hourly observed catchment-averaged sap flow and simulated sap flow in the growing season 2015 in the Weierbach catchment; b) hourly observed catchment-averaged sap flow and simulated sap flow in the growing season 2015 in the Huewelerbach catchment; orange points in a and b are simulations or observations within the dry period of July and August 2015.**



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
