# Peer review of "Leveraging sap flow data in a catchment-scale hybrid model to improve soil moisture and transpiration estimates"

_Hydrology and Earth System Sciences, 2022_

## Author Comment (AC1)

**Response to comments of Anonymous Referee #1 (R1)**

**R1:** *This manuscript details a study that makes use of sap flow measurements in a catchment-scale model, through the canopy resistance parametrization.*

*This is an interesting topic since (i) sap flow measurements are not common, particularly at the catchment-scale and probably as a consequence, (ii) the current parametrizations of the transpiration process in hydrological models are poorly constrained.*

*The paper is well written, the objectives are clearly stated and the results are nicely presented. I have only minor comments concerning mainly the metrics used to assess the differences of the model outputs, the conclusion on machine learning modeling that is in my opinion over-optimistic, and some methodological details that are not detailed enough.*

**Ralf Loritz (RL):** We thank Reviewer #1 (R1) very much for their positive assessment of our work and will address all their comments in a revised manuscript (MS).

Line by line comments:

**R1** (l.83-84): *why not use the Machine Learning (ML) model on conductance estimates instead of sap flow data? Could this improve the ability of the ML model to reproduce gc_sap ?*

**RL:** The performance differences between estimating canopy conductances ($g_c sap$) directly with the machine learning model or sap flow and afterwards calculating $g_c sap$ are minor. Adding this intermediate step shows, however, that sap flow (an independent observation) can be predicted by a recurrent neural network (RNN) and opens the option to calculate transpiration directly in case catchment averaged plant specific parameters are available (Line 325 - 330). Furthermore, it opens the possibility to validate the model on independent sap flow measurements in case there are new or additional measurements available. We believe that this intermediate step gives the approach additional value and will explain this better in a revised MS as well as record the root mean square error when estimating $g_c sap$ directly with an RNN.

**R1:** (l.160): *what is the spatial resolution of the LAI estimates? How many tiles are considered in the catchment?*

**RL:** We used the Visible Infrared Imaging Radiometer Suite (VIIRS) Leaf Area Index (LAI) product at a 8-day and 500 meter resolution (product name VNP15A2H; doi: 10.5067/VIIRS/VNP15A2H.0010). We extracted data for each pixel in the basin area of the Colpach catchment (70 pixels). We also filtered the data to only process high quality cloudless images. We will add this information to a revised MS.

**R1** (l.161-162): *The term "reference model" for the simulation using the Jarvis-Stewart model could be changed since it may suggest that this simulation is the closest to reality.*

**RL:** We will rephrase this term to "benchmark model" to avoid any confusion.

**R1** (l.188-190)*: There are missing details here. Why is it necessary to fill the gap in canopy conductance estimates? Why not just drop the concerned time steps? Why is it necessary to smooth the time series*

*with a three-hour window? Since no details are provided, the reader cannot figure out if these choices were a priori or a posteriori choices. Anyway, a justification is needed here.*

**RL:** The transpiration module of CATFLOW needs continues time series of canopy conductances for the entire simulation period (technical reasons). As the proposed method to estimate canopy conductances from sap flow works only under certain conditions (L 184 - 190) we needed to fill the gaps where these conditions were not met. We hence used the conductances estimate by the build-in stomata model of CATFLOW to fill the gaps. It is important to note that this happens (typically) only at time steps when there is little to no transpiration. The latter entails that in theory we could have also set the conductances at this time steps to a value close to zero. We will explain this in a revised MS.

**R1:** *Why is it necessary to smooth the time series with a three-hour window?*

**RL:** Particular in the morning and evening when the vapor pressure deficit is low the proposed approach is very sensitive to small changes of sap flow. As the variance of the sap flow measurements is highest during these periods the catchment averaged sap flow estimate can be a bit noisy. We hence decided to smooth the time series to avoid unrealistic jumps. Again, it is important to mention that this effect is particular relevant in the morning and evening hours when transpiration is low. We will explain this better in a revised MS.

**R1** (l.196): *Similar to the previous comment, why use a sequence length of 96 hours preceding the prediction time step. Was this value optimized or chosen a priori?*

**RL:** This is a hyperparameter of our RNN and was found by trial-and-error and is an optimized value. We trained the model on the growing season 2014 and tested different model realizations (hidden size, learning rate, sequence length, etc.) in the growing season 2016 (test data). Finally, we validated the model in the growing season 2015 (validation data set) without changing any hyperparameters. This is not well explained in the current MS and we will update this section accordingly.

**R1** l.225-227: *I found the description of the results very incomplete. Canopy conductances estimated by sap flow and Jarvis-Stewart are also very different in terms of variability since gc_sap presents much higher temporal fluctuations. The discussion focuses on bias and Spearman correlation but I think that alternative metrics might be used to provide a complete figure of the differences. Please consider using the Pearson correlation coefficient and the ratio of variance, and/or the KGE.*

**RL:** Good point. We will estimate the Pearson correlation and KGE (2012) between the two time series and discuss the results.

**R1** Figure 1: *This would be nice to add the streamflow time series since this is the only integrative measurement available. Also, a map showing the location of the available measurements of the catchment would help the reader to figure out whether the soil moisture probes are representative of the catchment. This could also be discussed in analyzing the results of Figure 1. I do not understand why transpiration rates are plotted only at the monthly time scale. Showing the high-frequency values would be valuable. Does the transpiration rate from sap flow are much more temporally variable compared to the simulated transpiration rate from the Jarvis-Stewart model? Is this why the conductance time series were smoothed by the 3-hour rolling mean?*

**RL:** We will add the simulated and observed hydrograph and the hourly transpiration rates to a supplement and discuss the results there. We will add a map to a revised MS that shows the locations of the sap flow sensors, the gauges, the soil moisture sensors and the catchment boundaries of the Colpach and the Weierbach.

**R1** (l.247): *"The Weierbach fell dry on 61 days (> 0.001 mm h-1) during the three-year record." I did not understand this sentence.*

**RL:** On 61 days there was close to no runoff observed in the Weierbach creek. We will rephrase this statement in our MS.

**R1** (l.264-265): *I had an opposite interpretation of the outcome of adding 15 randomly picked days of the dry period. To me, this is proof of the lack of robustness of the ML model and proof of its inability to extrapolate.*

**RL:** The "extrapolation" term was referring to the test in a different catchment where the RNN model predicts higher sap flows as observed in the Weierbach without any further tuning. Your comment is valid and we will make this clear in a revised MS.

**R1** (l.310-320): *In line with my previous comment, I found that the statements expressed in this paragraph are biased in favor of the ML model. I am not a great defender of complex and heavily parameterized models but in my opinion, ML models are also "complex and uncertain" and they suffer from overparametrization.*

**RL:** Agreed. We will revisit this section and try to be more objective. We will also add a short paragraph and discuss existing more physically-grounded approaches than the applied Steward-Jarvis approach to model stomatal conductance – and underpin why the data-driven/ hybrid approach presented in this study may be more accessible to hydrologist versus getting to deep in the plant ecophysiological modelling with its promises and dangers.

**R1** (l.324-326): *I think that this sentence should be placed in methodology to help understand why the ML model is used to estimate transpiration and not canopy conductance.*

**RL:** Good point, we will move it (see also answer to comment 1).

---

## Author Comment (AC2)

**Response to comments of Anonymous Referee #2 (R2)**

**R2:** *The authors present a catchment-scale hybrid model which is leveraged by sap flow data for more accurate hydrological simulations. The results showed that the hybrid model could lead to more realistic soil moisture estimates than the conventional Jarvis-Stewart equation, especially during drought conditions. The hybrid model predictions could match soil moisture and transpiration equally well as model runs using observed sap flow data and more importantly, hybrid model has good potential extrapolation beyond the study site. Such kind of hybrid model approaches which integrate machine learning methods and physical laws could open promising perspectives for more parsimonious process parametrizations.*

*These very interesting results have great potential to benefit the scientific community. With some minor clarification, this manuscript will be considered for publication.*

**Ralf Loritz (RL):** We thank Reviewer #2 (R2) very much for their positive assessment of our work and will address all their comments in a revised manuscript (MS).

Line by line comments:

**R2:** *I just have several specific questions. First of all, they didn't provide the cross-validation results. Secondly, did you also try the normal neural network, not the GRUs?*

**RL:** Good point. In a revised MS we will we present the cross validation results (see also answers to Reviewer #1).

We tried different combinations of artificial neural networks (ANN), gated recurrent networks (GRU) and long short-term memory networks (LSTM). Overall GRUs and LSTMs performed the best. As GRUs need less computational time and have slightly less weights, biases and no cell state, we used GRUs and not LSTMs. We will shortly explain this in a revised MS.

**R2** *(Line 129, 130): So, the 32 trees are evenly distributed in the catchment area? Could you show them on a map?*

**RL:** The sensors are not evenly distributed in the area. The field campaign was designed to capture **"***the typical hydro-pedological characteristics of the Colpach and the Weierbach.".* We will add a map to a revised MS that shows the locations of the sap flow sensors, the soil moisture sensors and the catchment boundaries of the Colpach and the Weierbach.

**R2** *(Line 196, 197): how many predictions time steps? Use 96 hours to predict next hour or next 2 hours? Why not 24h, 48h or 72h?*

**R2** *(Line 197~198): How did you prove the network which consists of four layers (input, two hidden, output) with 128 cell/hidden state is the most appropriate structure? Will the different dropout rate affect the results significantly, e.g., 5%, 15%, 20%?*

**RL:** Both are hyperparameters and where identified by trial and error. We trained the model on the growing season 2014 and tested different model realization (hidden size, learning rate, sequence length, etc.) as well as different tpyes of ANNs and RNNs in the growing season 2016 (test data). Finally, we validated the model in the growing season 2015 (validation data set) without changing any hyperparameter. We have not tried all options systematically and most likely, you could identify a

model setup that outperforms our model given the current split sampling. This is not well explained in the current MS and we will update this section accordingly.

**R2** (Line 260): *So, you're using data from 2014 and 2016 to train the deep learning model, while use the data of 2015 as the test dataset? Did you try cross validation and set 2014 or 2016 as the test dataset to see the results? Are there significant differences between different catchments and years? Could you show the data distribution, e.g., boxplot, of different years and catchments?*

**RL:** Import point. We state in our current MS: "*The latter was the reason to choose 2015 as test period and not 2016, which would have kept the chronological order and led to overall lower errors without bias.*" We will add the root mean square error for scenarios in which 2014 or 2016 would have been the validation data set.

**R2** (Section 2.2.4): *It seems that the machine learning model is set to point to sap flow directly? Why not just let machine learning model predict the conductance directly? You could also introduce constrains into the loss function by using equation 1 and 2 to constrain the training process.*

**RL:** Comment to Reviewer #1: "*The performance differences between estimating canopy conductances ($g_c sap$) directly with the machine learning model or sap flow and afterwards calculating $g_c sap$ are minor. Adding this intermediate step shows, however, that sap flow (an independent observation) can be predicted by a recurrent neural network (RNN) and opens the option to calculate transpiration directly in case catchment averaged plant specific parameters are available (Line 325 - 330). Furthermore, it opens the possibility to validate the model on independent sap flow measurements in case there are new or additional measurements available. We believe that this intermediate step gives the approach additional value and will explain this better in a revised MS as well as record the root mean square error when estimating $g_c sap$ directly with an RNN.*"

**R2** (Section 2.2.4): *I also suggest you should have a flow chart or schematic map for clearly clarifying the hybrid model. This could be more friendly to the readers.*

**RL:** We will consider adding a flow chart to a revised MS.

**R2** (Line 293, 294): *Could you further explain why the gcDL under- or overestimates on peaks? It seems that the model can't not capture the peak value very well? I think if you let the machine learning model predict the conductance directly with constrains from equation 1 and 2 into the loss function, this problem could be mitigated.*

**RL:** Predicting the canopy conductance directly only slightly improves the simulation results. But we agree that there is a potential to further improve our model results by a more systematic model choice or adding constrains to the model training. We will discuss this in a revised MS.